# DietSensor: Automatic Dietary Intake Measurement Using Mobile 3D Scanning Sensor for Diabetic Patients

**DOI:** 10.3390/s20123380

**Published:** 2020-06-15

**Authors:** Sepehr Makhsous, Mukund Bharadwaj, Benjamin E. Atkinson, Igor V. Novosselov, Alexander V. Mamishev

**Affiliations:** 1Sensors Energy and Automation Laboratory (SEAL), Department of Electrical and Computer Engineering, The University of Washington, Paul Allen Center, 185 E Stevens Way NE AE100R, Seattle, WA 98195, USA; mukundbj@uw.edu (M.B.); mamishev@uw.edu (A.V.M.); 2Department of Health Services, Box 357660, School of Public Health, The University of Washington, Seattle, WA 98195, USA; batkinson@auburn.wednet.edu; 3Department of Mechanical Engineering, The University of Washington, 3900 E Stevens Way NE, Seattle, WA 98195, USA; ivn@uw.edu

**Keywords:** dietary measurement, IR sensor, food image processing, food scanning, diabetes treatment assistance

## Abstract

Diabetes is a global epidemic that impacts millions of people every year. Enhanced dietary assessment techniques are critical for maintaining a healthy life for a diabetic patient. Moreover, hospitals must monitor their diabetic patients’ food intake to prescribe a certain amount of insulin. Malnutrition significantly increases patient mortality, the duration of the hospital stay, and, ultimately, medical costs. Currently, hospitals are not fully equipped to measure and track a patient’s nutritional intake, and the existing solutions require an extensive user input, which introduces a lot of human errors causing endocrinologists to overlook the measurement. This paper presents DietSensor, a wearable three-dimensional (3D) measurement system, which uses an over the counter 3D camera to assist the hospital personnel with measuring a patient’s nutritional intake. The structured environment of the hospital provides the opportunity to have access to the total nutritional data of any meal prepared in the kitchen as a cloud database. DietSensor uses the 3D scans and correlates them with the hospital kitchen database to calculate the exact consumed nutrition by the patient. The system was tested on twelve volunteers with no prior background or familiarity with the system. The overall calculated nutrition from the DietSensor phone application was compared with the outputs from the 24-h dietary recall (24HR) web application and MyFitnessPal phone application. The average absolute error on the collected data was 73%, 51%, and 33% for the 24HR, MyFitnessPal, and DietSensor systems, respectively.

## 1. Introduction

Diabetes is a global health issue, with an estimated 30.3 million people affected in the United States alone [1]. Diabetes is a disease characterized by the body’s impaired ability to produce or respond to insulin, resulting in abnormal metabolism of carbohydrates and elevated levels of glucose in the blood and urine. The associated conditions include cardiovascular diseases, musculoskeletal disorders, and cancers [2]. Enhanced dietary assessment techniques are critical for maintaining a healthy life for a diabetic patient.

Moreover, hospitals must monitor their diabetic patients’ food intake to prescribe the correct amounts of medication. Malnutrition is a long-standing issue for hospitalized patients, and 30% to 50% of patients are diagnosed with it nationwide. Malnutrition significantly increases patient mortality, the duration of the hospital stay, and, ultimately, medical costs [3]. Additionally, nursing and nutrition staff are only able to assess approximately 50% of meal trays due to the staffing levels and logistics involved in meal delivery and pickup (Harborview Medical Center internal data). This paper presents a novel method called the DietSensor system (shown in Figure 1), which uses three-dimensional (3D) scanning technology to measuring the volume of a food item and calculate the exact nutritional intake.

The DietSensor system uses automatic 3D reconstruction using a commercial off the shelf (COTS) depth camera and the existing nutritional database provided by the medical center’s kitchen to determine the actual amount of food consumed, faster and at a lower cost compared to similar solutions. The DietSensor smartphone application is used to capture the leftover food volume on a patient’s plate, and the volumetric calculation algorithm (VCA) is used to measure the nutritional data of the leftovers. Once the measurement is done, the DietSensor system uses the reported nutritional data provided by the medical center’s kitchen to subtract and measure the consumed amount to be reported to the endocrinologist. The process is further elaborated in the Section 3, findings are presented in the Section 4. 

## 2. Background

There has been much recent research done on imaging for automatic dietary monitoring. Recent developments in sensor technology have made image-based sensing non-intrusive and easy to use. An essential aspect of such systems is an accurate volumetric estimation of the food being scanned. Currently, these systems can be broadly split into three categories—3D imaging using depth sensors, 3D reconstruction using 2D image(s), and 3D reconstruction using projected light (structured light).

Depth sensors have the distinct advantage of being able to produce depth information without much additional direct processing. There are a variety of sensor technologies for this task, with time of flight (TOF) sensors being common. A popular TOF sensor currently in use in academia is the Kinect sensor V2 and has been used in a variety of fields, as shown in [4]. An example of the Kinect sensor used for food measurement is the measurement of egg volume with accuracy reaching 93% [5]. However, common issues with TOF sensors are the warm-up time and temperature compensation of the sensor, as shown in [6,7]. 

3D reconstruction using 2D images is achieved by a variety of methods, using a single image, multiple images from various poses, and stereo images [8,9,10,11]. These methods have the advantage of being low cost, and image processing with 2D images is a well-understood problem with relatively low hardware requirements. However, these methods have their drawbacks. They usually require system calibration, with most implementations using a known reference target such as a chessboard. They are also highly dependent on the image pose and content, especially methods using a single shot to estimate volume, making them susceptible to significant errors.

3D reconstruction using projected light tries to find a compromise between the previous two methods by projecting structured light onto the items and capturing 2D images, such as in [12,13,14]. The projection of structured light solves the problem of calibrating the image as well as the differences in textures in the different food items being captured. For low distance sensing, this results in more accurate 3D models with low computational costs.

Food volume estimation using 3D depth measurements, while having improved, is still mostly confined to well-controlled lab environments. One approach for volume estimation is Point2Volume, which uses View Synthesis to leverage deep learning for recreating 3D point clouds [15]. The implementation uses an Intel RealSense depth camera to capture partial point clouds and performs 3D reconstruction as well as volume estimation with the use of a deep learning network. The system obtained an average accuracy of 85% in the lab, with individual food items captured in a studio with a specialized chamber for 3D scanning. However, the deep learning network requires a large dataset for training, and the network does not perform well with objects not present in the training set. 

Another implementation using a mobile app uses a monocular Simultaneous Localization and Mapping (SLAM) system for volume estimation [16]. Data was captured using an iPhone 6 plus as well as a 4k wearable action camera for comparison. Volume estimation was done on food items in the lab with a black background and a Rubik’s cube for reference. With this setup, the system averaged a percentage accuracy of 83%. 

While these methods have been shown to improve accuracy, they have not been incorporated in hospitals, where they are most needed, in part due to a lack of testing in real-life situations. In both cases, the systems were tested in a controlled setting where the lighting, background, and other experimental parameters were fixed. They were also operated by the scientists who developed the system and not tested by an external human subject. Further, the food items tested were single items with uniform shapes, and each scan was made with the items kept separate from each other, representing a straightforward problem. Real-life usage typically requires scanning of irregular and complex shapes with multiple items together.

Another reason for the slow integration is the complexity and learning curve of a new solution. As the system complexity increases, more extensive user training is required. User training is an expensive and resource-intensive task and must be minimized to the extent possible. This paper aims to address the issues of real-life testing and system complexity using the DietSensor system. The DietSensor system is an incremental upgrade on previous monitoring solutions. However, it is redesigned to meet the needs of the hospital environment first, and by extension, the needs of a practical, real-life setting. The focus of the system is to reduce user burden while improving the accuracy of dietary intake estimation, especially the measurement of food items that are harder to quantify by a person visually.

In the hospital environment, the nutritional information of a meal must be estimated manually, either by taking a food journal/survey such as 24-h dietary recall (24HR) or manually weighing the meal and the leftovers by scale. Both techniques are extremely time-consuming [17]. Manual measurement is difficult to perform for all patients with the required accuracy; hence, the amount of insulin to be prescribed is estimated rather than exact.

24HR is a structured interview, intended to capture detailed information about all foods and beverages consumed by the respondent in the past 24 h; most commonly, from midnight to midnight the previous day. The questions are designed to have an open-ended response structure to respondents to provide a comprehensive and detailed report of all foods and beverages consumed [18].

The United States Department of Agriculture (USDA) Automated Multiple-Pass Method (AMPM) is a computerized method for collecting interviewer-administered 24HR information. An extension of the AMPM is the Automated Self-Administered 24-h (ASA24) system, which guides users on self-administering the AMPM online. The ASA24 respondent website guides the participant through the completion of a 24HR, using a dynamic user interface. It asks respondents to report eating occasions and times of consumption [19]. While visual aids are used as cues to judge portion sizes, these are dependent on the user’s perception and can cause over/underestimation.

Based on those specific questions, physicians provide prescriptions and formulate a personalized plan. However, many diabetic patients’ responses have inaccurate estimations of the amount of food eaten, with an above 20% error when estimating the size of the food portion [20]. Moreover, bias from gender and weight also contributes to the errors in self-reporting [21]. Hence, the 24HR’s potential inaccuracy in dietary estimation puts diabetic patients at risk and negatively influences their proper treatments. Despite its drawbacks, the 24HR method is still commonly practiced due to its low cost and little equipment requirements.

Smartphone applications (apps) such as MyFitnessPal provide a more accessible and practical implementation for monitoring nutritional data. MyFitnessPal is a goal-based food tracking app where the user is awarded points for logging foods into their diary. The main objective of the app is to assist users in programs of weight loss or weight gain. To track food through MyFitnessPal, the user first selects a mealtime to track: breakfast, lunch, dinner, or snack. Next, the user is prompted to find the tracked food and is offered options from a setlist. The user may search the database, scan a barcode, select a previously tracked food, or add a new food of their choosing. Here, the user has an extensive array of foods to select from, raising the possibility of choosing incorrectly. The last step for the user is to estimate the number of servings consumed. Similar to the 24HR method, users are shown visual aids as cues to estimate portion sizes and hence, suffer from the same inconsistency of user perception.

Looking at other popular apps on the iOS and Android stores for dietary intake monitoring, two stood out for having tools to input information by using the camera. Calorie Counter by Fatsecret is a popular app on the Android app store. It has an option to use the camera but only uses it for food recognition and not volume estimation. Similarly, Calorie Counter by Lose It! is another popular dietary estimation app. It has an inbuilt tool called ’SnapIt!’ which can analyze a picture and detect the food present in it. In both cases, the user has to input the serving size to the best of their knowledge, re-introducing the same errors as in MyFitnessPal.

While these apps are more accessible than current alternatives, they are not designed to meet the required accuracy for medical evaluation. They cannot be relied upon solely for critical monitoring of patients [22]. However, they are used as an estimate due to their ease of use and user-friendly interfaces. Accordingly, in the following sections, the DietSensor system has been compared to the currently used standard, the 24HR method, as well as the popularly used MyFitnessPal.

Unfortunately, an acceptable percentage error for carbohydrate ingestion is difficult to arrive at due to factors affecting how dietary carbohydrate is metabolized. Factors like gut bacteria, hormone levels, type of carbohydrate ingested, and whether the carbohydrate is ingested with other items like protein or fat vary from case to case. Arriving at such an error requires further research with scientifically acceptable methodologies. Due to complexity, user burden, and underreporting, the current digital-based food intake analyses have not been able to produce scientifically satisfactory results when tested in field-based studies [23].

The primary focus of the DietSensor system is to develop a methodology to reduce user burden and requirements for user training by introducing automation using 3D reconstruction. Once such a system is in mainstream use, further development with newer sensors and better-tuned algorithms can be easily integrated within the same framework to reduce the percentage of random and systemic error further.

## 3. Methodology

The DietSensor system looks to improve dietary measurement monitoring in hospitals by calculating the consumed nutritional data. Most US hospitals measure the exact amount of food before delivering it to the patient and record it to the facility’s nutritional database. Access to this database allows the DietSensor system to obtain the base nutritional data of the meal. Once the patient finishes eating, the nurse or the nutritional staff scans the leftovers using the DietSensor smartphone application. The application collects the depth data from a Commercial Off The Shelf (COTS) depth sensor called Structure Sensor (made by Occipital [24]). This process is further detailed in Section 3.1.

While newer mobile phones are starting to obtain the required hardware capabilities to perform 3D scans, native 3D sensors are still not widely prevalent, and different manufacturers have implemented different variations of this technology in an area that is still nascent. The Structure Sensor from Occipital is a well-established unit with easily adaptable solutions for both the iOS and Android operating systems, providing more consistent measurements across devices. The DietSensor system has been implemented using both an iPhone and an iPad. However, in most cases, an iPad was preferred due to the larger screen size allowing for more comfortable viewing and interaction with the generated mesh. Regardless of the hardware, 3D reconstruction to volume estimation required a spatially closed 3D mesh; the DietSensor system has addressed this by using a generalizable hole-filling algorithm, which applies during post-processing.

Post-processing of the 3D models is done using the VCA algorithm, which outputs the volume of each segment of the mesh by filling any missing spatial holes. The obtained volumes are then subtracted from the originally recorded nutritional data by the kitchen to calculate the exact amount of consumed nutrients by the patient. With this data, the endocrinologists can reliably prescribe and administer the required amount of insulin. The VCA algorithm is further detailed in Section 3.2.

The calculation to obtain nutritional data from the measured volume is described in Section 3.3, and finally, the testing schemes are described in Section 3.4.

### 3.1. Structure Sensor

The scanner hardware consists of a smartphone, an infrared (IR) projector, and a camera module to record depth information, shown in Figure 2. The IR projector module projects a known scattering matrix of infrared light onto an object’s surface.

The IR camera receives the distorted matrix, which is warped by the object and uses the information to calculate depth using the relative distance between the matrix’s vertices. The hardware requires a smartphone application to operate. Occipital provides an open-source algorithm for 3D reconstruction, which can be integrated into a smartphone application. During the scanning process, the user takes a live scan (video) of the plate from different points of view using the smartphone application. The application dynamically generates a 3D mesh using these frames and displays it on the screen. The system also uses the host device’s gyroscope to gather orientation data, useful for composing the 3D mesh. Mesh formation occurs by capturing the scene in focus from multiple angles, ideally capturing the edges on the bottom surface. Once the application determines that the mesh has been entirely generated, the mesh is ready to be uploaded to the server with the user’s confirmation. The process of scanning and uploading an object using the DietSensor application takes under two minutes; the average time spent by a patient for each meal is over 15 min (Harborview Medical Center internal data).

### 3.2. Volumetric Calculation Algorithm (VCA)

The 3D mesh data is obtained in the form of an object file containing the vertices that comprise the triangular mesh and a preview JPEG image taken by the camera during the scanning process. The obtained mesh has imperfections due to blind spots during image capture as well as the boundary between individual food items and the plate; hence, the volume cannot be calculated accurately. The data is post-processed to obtain the items of interest, eliminate the remaining regions of the scan, and fix any imperfections that are present. Finally, the volume is calculated on the processed mesh.

Figure 3 shows the flowchart of the VCA. From the user-generated mesh, the plate is first segmented and removed, and the resulting hole is filled. At this point, the user manually selects the individual food items, and this splits them into individualized volumes. When the mesh is divided in such a manner, it produces multiple smaller meshes with defects in them, primarily holes. These holes must be closed for the volume of the items to be accurately calculated. Each segment selected is closed by filling the holes generated, and the closed meshes are then used to calculate the volume for each item. Nutritional data is computed for each identified volume.

Most of the post-processing was done using Autodesk Meshmixer, a free 3D reconstruction tool [25]. This application was chosen over several other tools due to its simplicity to segment individual items in the scan and the ability to perform the volume calculation with high accuracy. While Meshmixer can also be used for filling holes, this step was done independently to obtain better flexibility in this process and choose an algorithm best suited for the type of mesh being handled. 

Segmentation of the plate, as well as the individual food items, were done within Meshmixer. After filling the generated holes, Meshmixer calculates the volume of the object. The post-processing time is typically between one to three minutes per food item; however, this process can be integrated into the main algorithm in the next version of the application (please see Section 6).

While hole filling is necessary for generating the volume, it can introduce errors due to the inexact production of the hole. Since hole-filling algorithms are used in a variety of situations, the requirements for such an algorithm can vary from application to application. The general criteria that such an algorithm should satisfy is given by Podolak and Rusinkiewicz, 2005:Produce a non-self-intersecting watertight mesh.Process arbitrary holes in complex meshes.Avoid changing, approximating, or re-sampling the original data away from the holes.Incorporate user-provided constraints to allow the selection of multiple topologically differing solutions.Process large scanned meshes with a running time proportional to the size of the holes, rather than that of the input mesh.

For the DietSensor use-case, the focus was on the first three requirements when searching for a suitable algorithm. The input mesh is expected to be of a reasonably consistent size, and user input was kept to a minimum. Thus, points four and five were not of high priority.

To help identify an appropriate algorithm based on the specific application requirements, currently available methods were reviewed with the help of [26]. Algorithms based on polygonal representation were looked at specifically, and the comparison table was used to select the method of choice. Based on the features considered and tabulated in the comparison, the technique described in [27] was selected. As described in the paper, the main steps for the algorithm are:Identify holes in the triangular mesh.Generate the initial patch mesh using the advanced front mesh (AFM) technique [28].Refine the patch mesh based on the Poisson equation.

The generated meshes were scanned and analyzed to identify the holes present in them. Typically, holes appeared at the region of intersection between two neighboring food items as well as the plate. The processing pipeline was developed to ensure that there was only a single hole formed at any given stage, enabling the algorithm to work effectively. Once the hole is identified, Figure 4 shows the algorithm used to fill it.

The AFM technique uses a simple, iterative approach to generate a new patch. The boundary vertices of the hole $v_{i}$ are identified, their edges $\left(e_{i-1,i},\;e_{i+1,i}\right)$, and the angle between the edges $\theta_{i}$, calculated. Starting with the vertex with the smallest θ, the algorithm proceeds to each adjacent vertex, adding new triangles (and vertices, if needed) to the mesh. After each addition, the vertices hidden by the newly formed triangle is removed from the boundary vertex set, and the new vertices are added to the set.

The position of the new vertices is primarily based on three rules, depending on $\theta_{i}$ as shown in Figure 5. When $\theta_{i}\leq 75{}^{\circ}$, the three vertices $\left(v_{i-1},\;v_{i},\;v_{i+1}\right)$ are connected directly to form the new triangle. When $75{}^{\circ}<\theta_{i}\leq 135{}^{\circ}$, a new vertex $v_{n}$ is added on the bisector of the edges $\left(e_{i-1,i},\;e_{i+1,i}\right)$ lying on the same plane formed by the two edges, with the average edge length of the edges. For $\theta_{i}>135{}^{\circ}$, two new vertices are added $\left(v_{n1},\;v_{n2}\right)$ on the trisector of the edges $\left(e_{i-1,i},\;e_{i+1,i}\right)$ with the average edge length of the edges.

However, the addition of new vertices given by the above three rules could cause a vertex to be added within an already defined triangle or very close to an already existing vertex. The former causes problems with mesh formation and the latter is inefficient and produces a more cluttered mesh. This problem is addressed by selecting a threshold, ∈, to form a radius around all existing vertices. When a new vertex is generated, the distances to the surrounding vertices are calculated. If any distance is less than ∈, an existing one replaces the vertex. An example of such a scenario is shown in Figure 6. The generated vertex $v_{n2}$ is within the threshold distance from an existing vertex $v_{e}$; hence, the triangle is formed with $v_{e}$. In the case that there are two or more possibilities, the selection which would result in the closest triangle to an equilateral triangle.

The generated patch is then refined to produce a more accurate filling. The refinement proposed, based on the Poisson equation, was not wholly implemented to reduce algorithm complexity. From analyzing some partially processed meshes, it was found that most holes produced from the mesh were planar due to the propensity of food lying flat on a plate or with insignificant vertical curvature. For this reason, a refinement of the vertices based on harmonic functions was done without performing the additional steps of solving for normal, triangle rotation and mesh reconstruction using the Poisson equation.

Given a triangle $T=(v_{1},\;v_{2},\;v_{3})$, the discrete harmonic function is used to find a basic piecewise mapping to minimize the Dirichlet energy given in Equation (1),
(1)$$E=\frac{1}{2}\underset{S_{T}}{\int }{\left\|grad_{T}f\right\|}^{2}$$
where $S_{T}$ refers to the surface of triangle T, $grad_{T}$ refers to the gradient vector of the vertex within T, and f refers to an unknown scalar function.

The harmonic function is applied to a set of vertices called the 1-ring vertices surrounding any given vertex in the generated mesh. 1-ring vertices are defined as the set of vertices directly adjacent to the vertex being examined. For the vertex $v_{i}$, shown in Figure 7, the discrete Laplacian operator is given by Equation (2),
(2)$$\Delta f\left(v_{i}\right)=\underset{v_{j}\in N_{i}}{\sum }\frac{1}{2}\left(\cot\alpha_{i,j}+\cot\beta_{i,j}\right)\left(f_{i}-f_{j}\right)$$
where $f_{i}$ is one of the coordinates of $v_{i}$, $\left(\alpha_{i,j},\;\beta_{i,j}\right)$ are the two angles opposite to the edge $e_{j,i}$, and $N_{i}$ is the set of 1-ring vertices of $v_{i}$.

Therefore, it can be shown that the minimization problem can be expressed as (3),
(3)$$\begin{array}{cc} \sum_{v_{j}\in N_{i}}\omega_{i,j}\left(f\left(v_{j}\right)-f\left(v_{i}\right)\right)=0, & v_{i}\;\epsilon \;V_{I} \end{array}$$
where $\omega_{i,j}=\cot\alpha_{i,j}+\cot\beta_{i,\;j}$. The newer vertices are generated by solving for this linear system three times for each component (x, y, z), and the mesh is modified. As noted in the paper, by only performing this step, the resulting mesh is depressed and does not work well with items with larger curvatures. Modifying the algorithm for larger curvatures is further discussed in Section 6. 

To demonstrate the result of the post-processing, see Figure 8, which shows the hole-filling process done on the capture of a croissant and apple done together, with the post-processed apple after segmentation.

### 3.3. Nutritional Data Calculations

The volume data of the leftover meal is then converted into nutritional data using available databases. Food density data were obtained from the Nutrition Coordination Center (NCC) [29], and the volume is converted to its estimated weight given by (4):(4)W=V∗D
where *W* is the weight of the meal, *V* is the measured volume of the meal, and *D* is the density factor from the NCC database.

The estimated weight is then converted to nutritional data by looking up the nutritional information per gram of a given food item. The nutritional content per gram is multiplied by the weight of the food item found in the system, generating estimated nutritional information for each participant’s reported consumption, as shown in Equation (5),
(5)$$CN\;\left(i\right)=W*\;\frac{ND\;}{g}\left(i\right)$$
where *CN* is calculated nutrient, *ND* is nutritional data (*i*)—which refers to each dietary component measured, such as calories or fat—*W* is the weight, and g is for gram

The calculated nutrient is then subtracted by the initially reported amount by the facility’s kitchen using Equation (6),
(6)CoN=RN−CN (i)
where *CoN* is consumed nutrient, *RN* is reported nutrient, which is provided by the kitchen, and *CN* is the calculated nutrient of the leftover meal.

The consumed nutrient is then uploaded to the patient’s file to be accessed by the physician for an accurate prescription and overall diagnosis. 

### 3.4. Preliminary Studies

When considering a typical meal on a tray found in a health institute, sides are placed in separate containers on the tray, apart from the main plate. These containers are simple geometric shapes, differing from the amorphous and complex shapes produced by mixed meals. The DietScanner system was initially tested in the lab by estimating scans of geometric shapes, further described in Section 3.4.1. 

The next round of testing in the lab was with model food items, simulating examples of whole foods placed on a tray, such as a fruit. These model food items were made of plastic and consisted of an apple, a croissant, a pear, and a pepper. Volume measurement was performed multiple times, and the errors averaged. Testing with model food items is further described in Section 3.4.2.

The final round of testing involved scanning multifood plates for measuring the volume of prepared plates of food, simulating the actual measurement of a plate with complex food items by a user. The test is further described in Section 3.4.3.

#### 3.4.1. Known Geometric Shapes

The first stage of testing involved scanning a known cube and cone as objects with a well-defined volume. The cube represented an object with prominent edges, while the cone was used to test the implementation on smooth surfaces. These objects were tested resting on a flat plate and were scanned individually.

Simple geometric shapes such as these rendered well, and the calculated volumes were accurate. For example, for the 7 cm cube shown in Figure 9, the calculated volume after post-processing was 346.55 cm^3^ compared to the actual volume of 343 cm^3^. These scans showed improvements in overall measurement accuracy; during the preliminary testing with regular shapes, the average accuracy was 95%.

#### 3.4.2. Model Food Items

Similar to the previous test, model food items (apple, pear, croissant, and pepper) made of plastic were used to validate the scanner’s accuracy. These were scanned individually as well as with multiple models on a plate. The average error amongst twenty-four scans of these models was 5%, with specific models being measured better than others. For example, the apple model had a mean error of 7.5%, while the model of the pepper had a much lower mean error of 2%. This accuracy is comparable to other publications in this field, tested in a controlled environment, with a trained user operating the system.

Individual model error was dependent on the type and placement of the model. For example, the apple model had the highest error due to the incorrect estimation of the bottom surface by the VCA. Unlike simple geometric shapes, irregular surfaces were harder for the algorithm to fill and resulted in more significant discrepancies. Figure 10 shows a 3D scan of apple and croissant models (plastic food) in mesh formation after post-processing with the VCA. The models are placed on a flat surface with both items touching each other. The resulting scan captured the pose in detail, with surface details rendered well. However, during post-processing, the two models had to be split for individual volume estimation. Running the VCA resulted in a volumetric error of 5% for the croissant and 14% for the apple.

#### 3.4.3. Multifood Plates

The next evaluation of the DietSensor system was to determine its capability of measuring the volume of complex food items. This was tested by generating scans of prepared plates of food. For the purpose of evaluation, a plate of mashed potatoes and meatloaf was scanned. An example of a scanned meal produced by DietSensor is shown in Figure 11. Figure 11a shows a captured scan of a plate of meatloaf and mashed potatoes.

The scanned plate was first segmented to remove the plate and utensils from the food. The segmented mesh has the meatloaf and mashed potatoes captured together, This is now run through the hole filling algorithm to fill the holes present, as shown in Figure 11b. Now, from this joint mesh, the meatloaf and mashed potatoes are segmented separately. Each segment is shown in Figure 11b by the perimeter drawn around them. The meatloaf has a perimeter in blue and the mashed potatoes in orange. In most cases, the food items were well-segmented from the captured scan. Hole filling generated meshes of sufficient quality to represent the lost volume. Depending on the quality of capture, the larger holes between the food items and the plate varied in size, with better scans minimizing the hole.

The test was also extended to capture real-life usage. The test was carried out with three different food plates scanned by test participants, and the results are presented in Section 4.

## 4. Results

The DietSensor system was tested in a real-world scenario and simultaneously compared with the 24HR method using the ASA24 Dietary Assessment Tool and the MyFitnessPal application to study the overall system accuracy, performance, and ease of usability. The system calculates standard nutritional metrics such as calories, carbohydrates, protein, sugar, and fats. However, since the primary application of the system is for diabetic patients in hospitals, the metric focused on in the results is carbohydrates. Nutritional information was obtained from a collection of sites, as the spread of food items in the experiment were not all found on a single nutritional database. Both SelfNutritionData [30] and FatSecret [31] are used to cover the entire scope of the foods tested. Both have nutritional data available in 100 g portions, allowing for easy conversion to per gram data. 

Twelve randomly selected participants volunteered to consume a prepared one of the prepared plates of food described in the previous section and use the three dietary tracking methods to record their nutritional intake. Three different plates of food were prepared for testing, with each plate contained one to three food items, with different arrangements, including food items in-contact and separated (see Figure 12). There were three recipes prepared for the study, which were randomly matched to volunteers (while considering any allergies). The three plate configurations were (Plate 1) chicken breast and vegetables, (Plate 2) garlic roasted vegetable pasta and, (Plate 3) meatloaf and mashed potatoes.

The participants were split into three different test groups with varying levels of pre-known information about the meal to identify the robustness of the system to user error and experience. Group A was given the exact recipe for the prepared food, Group B was given a list of recipes to pick one they considered most accurate, and Group C was served a plate with no information. The accuracy of the system was compared across the three groups to identify how well the system worked with user input with discrepancies. Group C users were not made to use MyFitnessPal (under the assumption that they are not aware of any information about the plate other than what they could visually see). The participants then followed these seven steps to measure their intake:Fill out the initial questions in the 24HR dietary recall survey.Fill out the initial questions in the DietSensor survey.Scan plate using the DietSensor scanner.Eat as much of the meal as they wanted.Complete the 24HR and DietSensor Surveys based on leftovers.Rescan leftovers with DietSensor scanner.Fill out the MyFitnessPal application.

The control measurement was made with the use of a weighing scale and manual measurement tools (i.e., measuring cups). Each plate of food was prepared for the experiment and manually measured before being taken by the tester. The plate was then re-weighed after the tester finished eating (along with any leftovers). The net difference in weight provided the control measurement against which all other methods were compared.

Both the ASA24 and MyFitnessPal were completed by human subjects. Each participant followed the instructions presented on the screen by the website and app, respectively. However, the 24HR method is typically done for the entirety of the previous day; only the meal eaten was evaluated by this method for the purpose of this study.

The absolute error was determined as a better metric than percentage error per plate to evaluate the methods due to medication being prescribed as a function of overall nutritional intake, independent of the actual size of the meal. Table 1 demonstrates that the DietSensor system had a much smaller absolute error with a lower standard deviation (SD) compared to the other two methods, thereby giving a better indication of dietary intake. The 24HR method performed better than MyFitnessPal, although both had a considerably higher mean absolute error and standard deviation. Overall, the DietSensor system achieved an absolute error of 255 g (33%) with a standard deviation of 14 g, the 24HR method achieved an absolute error of 389 g (51%) with a standard deviation of 34 g, and MyFitnessPal achieved an absolute error of 390 g (73%) with a standard deviation of 28 g.

The obtained results were compared against some of the results previously reported using the 24HR method. From the results shown in [20], the error obtained from the group of users labeled as “three-dimensional unguided in-person interview groups” (3DUIP) was used for comparison. The paper presented an average of all measurement errors from the test, averaging both under- and over-estimated values. Averaging of signed error means that if a respondent over-estimated his consumption while another under-estimated it, the average misestimation error remains close to zero, and on the whole, reduces the overall mean system error.

For foods considered ’mounded’ or amorphous, the average error for the 3DUIP group varied with the food eaten. Green beans were underestimated by 42% (Inter Quartile Range (IQR) of ~10%), salad, and french fries by ~24% (IQR of 20%), and macaroni and cheese by 8% (IQR of ~30%). While IQR and SD are not interchangeable, they provide an understanding of the variation in reported error. For example, while respondents estimated macaroni and cheese to within an 8% mean error, the IQR of 30% represents the wide range of errors collected. This range of errors was similar to the errors obtained using the ASA24 to perform the 24HR method to compare to the DietSensor system. The results were also split by the user group and the results tabulated in Table 2.

For the DietSensor system, the average absolute error increased from 7 g to 17 g and 28 g for groups A, B, and C, respectively. Similarly, for the 24HR method, error increased from 30 g to 31 g and 42g. Apart from the lower absolute error, the DietSensor system performed more uniformly across the three plates. MyFitnessPal was only performed by groups A and B as Group C were not told what they were eating, making it difficult to enter information into the application accurately. MyFitnessPal had the highest error amongst Group A users with an average error of 74 g but performed well amongst Group B users with a mean error of 26 g and a distribution very similar to the DietSensor system.

## 5. Discussion

The two-fold approach of the DietSensor system was effective in increasing the accuracy of current methods in dietary tracking. The scanner was useful for obtaining accurate volumetric estimations through 3D reconstruction and increased the accuracy of the nutritional data filled by the medical staff. 

The DietSensor system was tested in a controlled environment, and the results obtained were compared against the existing state of the art solutions. While these solutions could not be tested in the lab alongside DietSensor, available data from their papers was used instead. Testing was mainly focussed on the key focus of the DietSensor system—its efficacy in a semi-controlled situation with minimal user burden. The DietSensor system was therefore compared to the prominent solutions in use at the institute, the 24HR method, and MyFitnessPal.

The 24HR method has been shown to reasonably estimate dietary intake with studies showing mean misestimation within 20% [20]. However, in a study such as this, participants are interviewed by a trained interviewer. The interviewer had previous training in cognitive interviewing skills, had been trained to use the USDA three-pass method (now expanded to a multiple-pass method) for 24-h recalls, and had conducted dietary recalls previously [20]. The results obtained in our testing were obtained from users who had no prior experience with the 24HR method and were tested using the automated ASA24 tool with limited directions; this was indicative of our final proposed solution of an accurate, self-administrable solution. Further, these studies also specifically mention difficulty with foods classified as ‘amorphous’ and the obtaining of better results with foods classified as ‘single-unit’ [20,32,33], with ‘amorphous’ foods having much higher errors than the reported overall mean. From this classification, it is observed that apart from the chicken breast served in Plate 1, all other food items used for testing belonged to the ‘amorphous’ category in our testing. The high number of ‘amorphous’ food items also corresponds to the higher mean errors observed from the 24HR and MyFitnessPal methods as well.

Volume estimation performed with the DietSensor on individual food items in a controlled setting achieved an average accuracy of 95% when capturing individual items. From the results obtained from user testing, the DietSensor system had an overall lower absolute error compared to the 24HR method and MyFitnessPal. Significantly, it was also consistently achieved a smaller error, as indicated by the low standard deviation. The system also produced low error with all three user groups; while Group C users produced higher errors compared to the other two groups, the range and mean of the error was still within the results obtained from the 24HR method.

Many publications on image-based volume estimation have been tested in a controlled environment, with a trained user operating the system. A controlled environment encompasses variables such as background, lighting, plate size and shape, food type, and plating. The movement from a completely controlled environment, such as the lab to a semi-controlled environment such as a health institution, introduces new challenges, with variations in the factors mentioned above. Identifying these factors is the first step in rolling out such a system in a completely uncontrolled environment, such as for domestic users. Additionally, lab testings are predominantly performed by the scientists who developed the system, and not tested by an external human subject. When a system is used outside the lab, chances of user error are higher; as system complexity increases, more extensive user training is required. User training is an expensive and resource-intensive task and must be minimized to the extent possible. Reducing user burden (making the system less complex to operate) is an important factor for a system to succeed commercially, and balancing higher accuracy with user burden is critical.

From the results, the difference in average accuracy between a lab-based result and field-based testing highlights the complexities introduced by different environmental conditions, food type, and user training and their ability to scan the meal properly. The DietSensor system is developed to consider these challenges to minimize the resulting error and a 33% error rate while reducing user burden shows promise.

For the design characteristics of a system which is designed for a hospital environment there are other external factors in play that must be considered to keep the system accurate. By collecting and analyzing usage data from staff and patients inside the hospital, common measurement errors and usage patterns may emerge, leading to modifications that can allow the system to perform well in external conditions. For this purpose, the possibility of performing a beta study at the Harborview Medical Center is being explored. The DietSensor system would be integrated with the current system in place, and the results evaluated.

Finally, it must be noted that while the targeted audience is diabetic patients, a problem faced by health institutions is malnutrition amongst their patients. The system, therefore, in the current form, can help improve the nutrition analysis of all hospitalized patients.

## 6. Future Work

Further study in utilizing different mesh fixing techniques, depending on the mesh, would help reduce errors and make the system more robust and autonomous. Newer algorithms suggest that it may be possible to get better results by distinguishing between different hole sizes and performing the appropriate filling method as proposed in [34]. While hole size alone might not be enough for automatic mesh fixing, additional characteristics of the mesh and boundary may increase its efficacy. These include characteristics such as the hole type (simple/complex), as well as some known information about the item. Development in making the scanning process more autonomous would also improve the usability and accuracy of the system, with the added benefit of solving some of the problems faced during mesh formation. One such method would be to find the means to automate the segmentation process. Algorithms to separate food from the plate as well as each other automatically would increase the accuracy and usability of the system. 

Complete system automation would entail the above steps as well as automating the Meshmixer processing to make the system entirely devoid of human input except for edge cases. Such a system also opens the possibility of the system being modified to be directly used by patients with minimal training instead of by medical/nutritional staff. The ability of patients to monitor their food nutrition would relieve some burden from the medical team while maintaining the requisite quality for proper treatment.

The promise of newer mobile phones coming with a built-in 3D sensor is positive for a system such as this. Modern flagships such as the Galaxy Note 10+ and the new iPad Pro (2020) are already equipped with depth sensing scanners [35,36]. Further, the availability of a ToF sensor on a mid-tier phone, such as the Honor Vision 20 [37], is a sign that the technology is trickling down to lower-powered devices. While the Structure Sensor is a single addition and can remain compact, a sensor embedded in the phone makes the setup easier to deploy.

## 7. Conclusions

There is a clear need to move towards automated dietary measurement for enhanced treatment of diabetes. Currently, apart from manual measurement, dietary intake is estimated by using an automated 24HR implementation such as the ASA24 or popularly available applications such as MyFitnessPal. While there are many developments in automated measurement, these have not reached widespread implementation due to the constraints of accuracy, cost, and system complexity. This paper presented a system capable of determining the precise amount of food consumed in a hospital setting. The DietScan system proves that it is possible to integrate accurate measurement using 3D imaging without significant disruptions to the current operating pipeline of a hospital. The use of a straightforward interface, inexpensive equipment, and portable form-factor paves the way for modifying the pipeline by allowing users to participate in the process, relieving the medical staff of some of the load.

The DietScan system was tested on a small group of participants, and the experimental results were compared to values obtained from using the ASA24 and MyFitnessPal applications. The tests show that the DietSensor system can achieve better accuracy while reducing user burden, and remaining more consistent with discrepancies in input details. The time taken to run the system of approximately two minutes is less than the average time taken of 15 min per meal, allowing it to be operated in real-time without a discernable lag. 

## Figures and Tables

**Figure 1 sensors-20-03380-f001:**
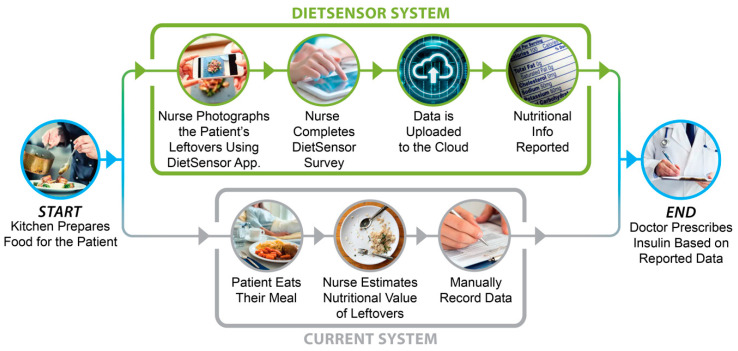
The DietSensor system replaces the existing estimation process by introducing an automated smartphone application to measure the leftover food volume on a patient’s plate. The DietSensor system subtracts the reported nutritional data provided by the medical center’s kitchen to measure the consumed amount and report to the physician.

**Figure 2 sensors-20-03380-f002:**
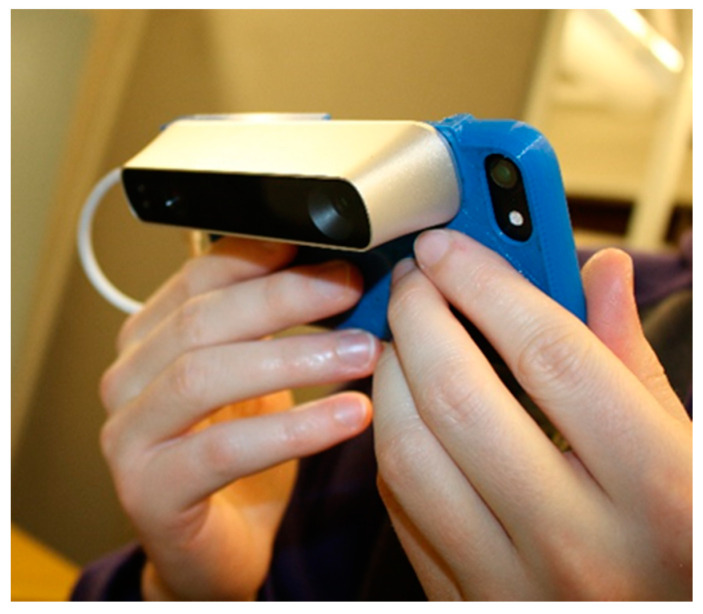
A user is holding an iPhone with the Structure Sensor.

**Figure 3 sensors-20-03380-f003:**
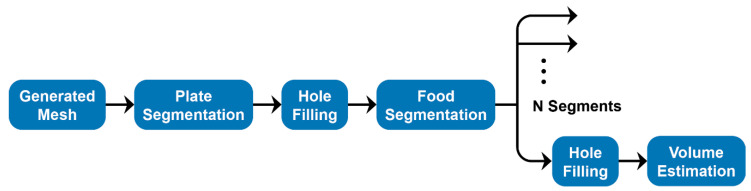
Flowchart of the volumetric calculation algorithm (VCA). Apart from Hole Filling, all steps were performed on MeshMixer. The generated mesh is passed as input to the pipeline, and the calculated volumes of each food segment are sent as output. Plate and food segmentation were done separately to reduce errors and ensure only a single hole was present at any stage of the pipeline. The hole-filling algorithm is explored in more detail below.

**Figure 4 sensors-20-03380-f004:**
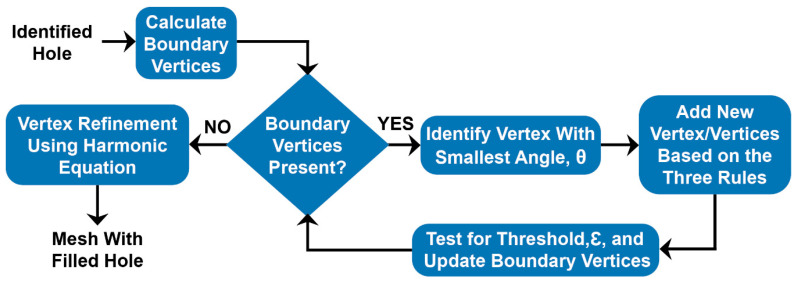
Algorithm flowchart for hole filling. Each identified hole is run through this process to be filled. For each hole, the boundary vertices are found, and the advanced front mesh (AFM) technique is used to fill the hole. New vertex/vertices are added based on the rules detailed below.

**Figure 5 sensors-20-03380-f005:**
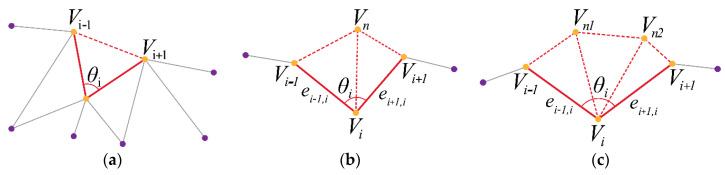
The three rules for the new vertex generation, depending on vertex angle $\theta_{i}$: (**a**) for $\theta_{i}\leq 75{}^{\circ}$, (**b**) for $75{}^{\circ}<\theta_{i}\leq 135{}^{\circ}$, and (**c**) for $\theta_{i}>135{}^{\circ}$. $v_{i}$ refers to the vertex in focus with the associated angle $\theta_{i}$, $\left(v_{i-1},\;v_{i+1}\right)$ refer to the two adjacent vertices, $\left(e_{i-1,i},\;e_{i+1,i}\right)$ refer to the two edges connecting the adjacent vertices to $v_{i}$, and $v_{n}\;,(v_{n1},\;v_{n2})$ refer to the newly added vertices. These vertices were generated in the holes produced during DietSensor post-processing to achieve watertight meshes.

**Figure 6 sensors-20-03380-f006:**
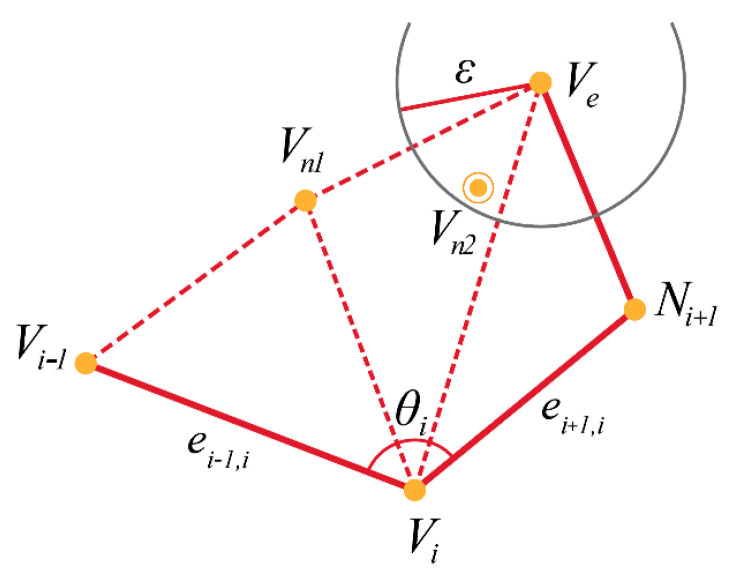
The threshold to check for overlaps and reduce mesh complexity where $v_{e}$ is an existing vertex within a radius of ∈ to the proposed vertex $v_{n2}$. For the DietSensor application, larger values of ∈ were experimented with to simplify the mesh as much as possible to be computationally effective while retaining the general 3D geometry.

**Figure 7 sensors-20-03380-f007:**
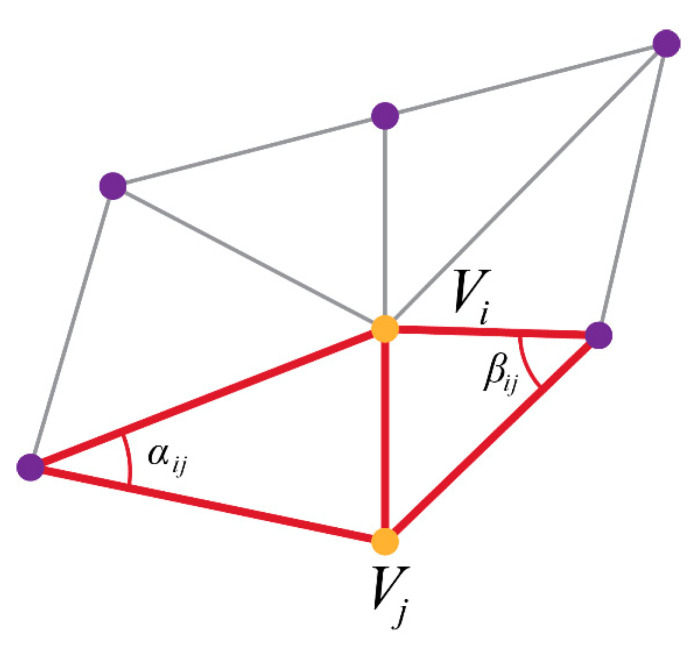
Angles and vertices for harmonic-based vertex calculation. The angles $\left(\alpha_{i,j},\;\beta_{i,j}\right)$ are used to calculate the gradient between vertices.

**Figure 8 sensors-20-03380-f008:**
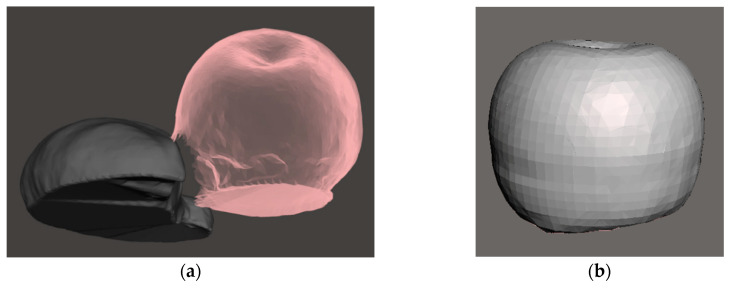
A 3D rendered mesh can produce significant errors if not handled well: (**a**) shows an apple and croissant captured together, which creates a hole during segmentation; (**b**) shows the corrected mesh for the apple after hole filling (14% error of underreporting in this specific case).

**Figure 9 sensors-20-03380-f009:**
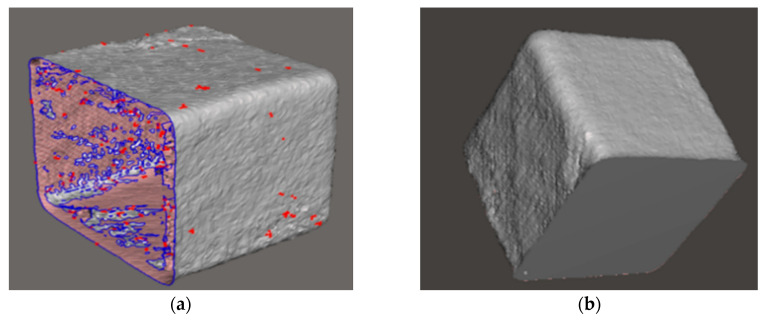
This figure shows the raw 3D mesh of a cube (**a**) with an unclosed mesh on one side. Once the raw 3D mesh data is passed through the post-processing algorithm, all holes are closed (**b**), and volume is calculated. The volume error of the post-processed cube is under 5%.

**Figure 10 sensors-20-03380-f010:**
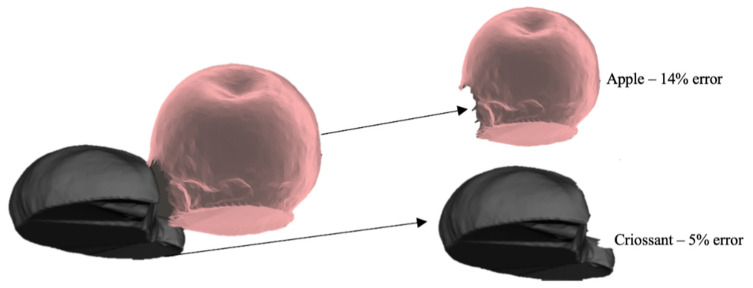
A model croissant and apple in contact with each other and scanned to produce a well-captured surface mesh as well as display the boundary between them.

**Figure 11 sensors-20-03380-f011:**
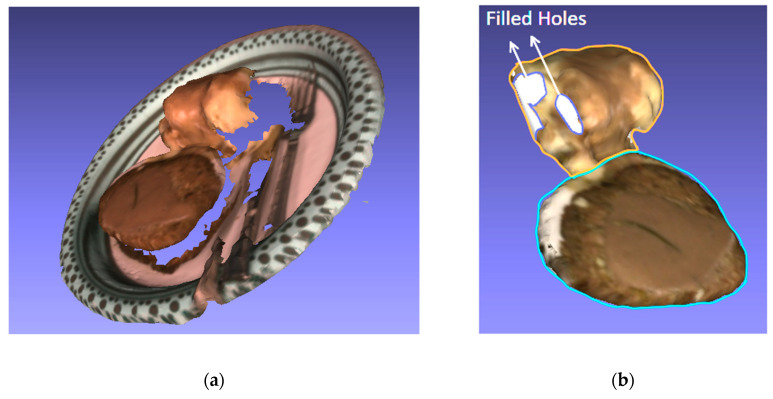
Example of a captured surface mesh of a plate of meatloaf and mashed potatoes: (**a**) shows the initial scan before preprocessing, including the meal, utensils, and plate; (**b**) shows the segmented scan with the plate and utensils removed and hole closing performed. The perimeter of each segment is highlighted (meatloaf in blue, mashed potatoes in orange).

**Figure 12 sensors-20-03380-f012:**
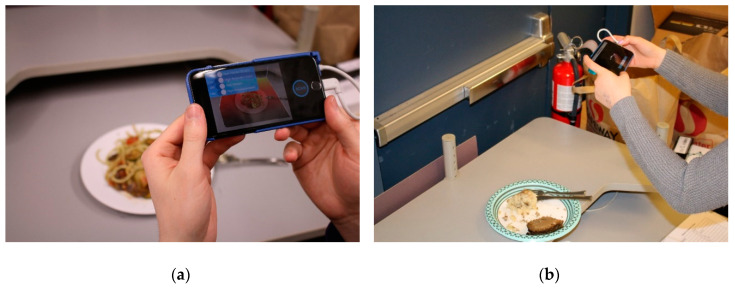
Participants using the DietSensor system during the comprehensive system testing with the prepared food. (**a**) A participant using DietSensor before eating, of Plate 2 with mixed food items, and (**b**) a participant scanning the leftovers of Plate 3 with food items separate.

**Table 1 sensors-20-03380-t001:** The actual volume for each meal, along with the observed and absolute error for each method, in grams (g). For each trial, cells highlighted in green are the best result (lowest error) out of the three methods (DietSensor, 24HR, and MyFitnessPal). Cells highlighted in yellow are results that were not the best but still had an error of less than 10 g, and cells highlighted in red were results with errors above 10 g and did not have the lowest error amongst the three methods. These results contain data from all three groups (A, B, and C). Group C users were not made to use MyFitnessPal (under the assumption that they are not aware of any information about the plate). These cells have been labeled NF (Not Found).

Actual Value (g)			24HR (g)		MyFitnessPal (g)		DietSensor (g)	
			Estimated	Error	Estimated	Error	Measured	Error
19			34	15	78	59	30	11
100			42	−58	52	−48	122	22
30			39	9	NF	NF	80	50
121			46	−75	NF	NF	94	−27
18			59	41	NF	NF	25	7
61			58	−3	156	95	62	1
46			49	3	68	22	39	−7
78			75	−3	56	−22	39	−39
61			44	−17	NF	NF	28	−33
86			24	−62	17	−69	104	18
14			129	115	NF	NF	7	−7
58			26	−32	50	−8	35	−23
84			13	−71	17	−67	94	10
**Absolute Error (g)**				389		390		255
**Absolute Error (%)**				51%		73%		33%
**Standard Deviation (g)**				34		28		14
green	yellow	red	grey					

**Table 2 sensors-20-03380-t002:** A subset of data from Table 1 taken to highlight variation in absolute error in grams (g) between each method, depending on the user groups (A, B, and C). Group C users were not made to use MyFitnessPal (under the assumption that they are not aware of any information about the plate). These cells have been labeled NF (Not Found). The rows are marked by the plate consumed, and the values are color-coded with lower errors colored dark green, and larger errors colored dark red.

Method		Group A		Group B		Group C		Plate	
24HR		15		59		41		1	
		3		3		9		2	
		71		32		75		3	
MyFitnessPal		59		48		NF		1	
		95		22		NF		2	
		67		8		NF		3	
DietSensor		11		22		7		1	
		1		7		50		2	
		10		23		27		3	

dark green					dark red				

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
