# Peer review of "DietSensor: Automatic Dietary Intake Measurement Using Mobile 3D Scanning Sensor for Diabetic Patients"

_sensors, 2020, doi:10.3390/s20123380_

Round 1

Reviewer 1 Report

The paper addresses an important area, which is the measurement/assessment of dietary intake. For this purpose, they developed a 3D scanning sensor (named DietSensor). The idea is to scan the remaining food on the plates, calculate their volume, and compare it with the baseline meal ratios. The idea has some merits and compared to previously used estimations based on 2D images or TOF camera, this solution also deserves to be published. However, I have several concerns which need to be addressed first:

  • The performance of the DietSensor should be validated quantitatively in more detail. 3D reconstructions are only shown for simple cases/test scenarios. It would be important to see how the software reconstructs a more complex dish e.g. main dish + side dishes (meat + pasta/vegetables or side dishes with an amorphous surface like rice/mashed potatoes, pasta, etc.). I think the authors should form rational food categories/types and test the software’s performance on them. E.g. mashed potato, measure it 10 times after various portions remaining, control the measurements by weighting with a scale and calculate the error. It would be important to see how the software performs on different food types/categories.
  • The readers should not be considered familiar with either the 24HR or the MyFitnessPal application. The exact methods of how the remaining food is given in these systems should be described.
  • The use of two applications cannot be considered ‘measurements’, please do not refer to them as such (e.g. Table 1). Based on the reviewer’s experience with such applications, I think the user needs to estimate the size of the remaining dish and input them by selecting pre-defined dishes/ratios. The whole procedure is far from measurement, this is merely a rough estimation by the user.
  • From this point of view the only measurement is the authors' DietSensor solution. The real advantage/benefits of this solution could be judged compared to other measurement (not estimation) methods. If the authors have access to any such method (listed in the introduction), a more meaningful benchmarking could be done.
  • In the introduction the authors mention that “However, many diabetic patients’ responses have inaccurate estimations of the amount of food eaten, with an above 20% error when estimating the size of the food portion [16].” Here the precision of the DietSensor was 33%, which is worse than this reference. What would be a realistic goal?
  • Newer mobile phones have both the hardware and software capabilities to perform 3D scans all alone. There are several applications available for this. How do they compare to the authors’ solution? Wouldn’t it be possible to use a phone alone with any additional hardware for the same purpose?

Reviewer 2 Report

Overall good job working on the development of a system to determine the amount of food. 

Comments:

  1. The methods (i.e. 24HR and MyFitnessPal) used to compare the presented method (i.e. DietSensor systems) don't seem to have the same structure, there are many published papers that present similar image processing methods that have lower error than the presented method and have not been mentioned in this study. The introduction should be improved to highlight other methods and why the method presented in this article is considered a contribution.
  2. What percentage of error is acceptable, when considered diabetic patients?
  3. Also if this system functions well in a health institute, most diabetic patients are not present in these institutes, how would you consider improving the system taking this into account.
  4. The text needs to be more clear when presenting the results, instead of just having the error quantity, the standard deviation should also be included as a quantity in the text.
  5. There are many publications on image processed food intake analysis that should have been at least mentioned in the introduction of this paper and therefore this paper should undergo further steps to justify their contribution.

Round 2

Reviewer 1 Report

I reckon that the authors made a great effort to correct their manuscript. The quality of the revised/extended version improved significantly. I can thus support the publication of the paper in the journal. Thank you for your work.

Author Response

The authors would like to thank the reviewer for the great comments, feedback, and suggestions. We are happy that our changes were satisfactory and addressed your concerns. 

Reviewer 2 Report

Great work on improving the manuscript. Minor comments:

Line 478 - the SD for the 24HR method should be 34g instead of 14g?

Recommendation - I would move some content from the results section to the methodology section, I noticed some of the material in the results section is mainly about methods.

Author Response

The authors would like to thank the reviewer for his great comments, feedback, and suggestions. We are happy that our changes were satisfactory and addressed all your concerns. 

We have addressed your comment and recommendation and the changes are highlighted in green in the new revised version of the manuscript.

  • Line 478 - the SD for the 24HR method should be 34g instead of 14g?
    • Thank you for pointing this out, we have addressed this error and shown in green font - line 493.
  • Recommendation - I would move some content from the results section to the methodology section, I noticed some of the material in the results section is mainly about methods.
    • Thank you for this recommendation, we have moved some of the content from the Results section to the Methodology section. The changes are shown from line 370 to 438. Please see attached for a more detailed response letter which includes the text itself.
